# Controlled and Sequential Delivery of Stromal Derived Factor-1 α (SDF-1α) and Magnesium Ions from Bifunctional Hydrogel for Bone Regeneration

**DOI:** 10.3390/polym14142872

**Published:** 2022-07-15

**Authors:** Zhengshi Li, Huimin Lin, Shanwei Shi, Kai Su, Guangsen Zheng, Siyong Gao, Xuan Zeng, Honglong Ning, Meng Yu, Xiang Li, Guiqing Liao

**Affiliations:** 1Department of Oral and Maxillofacial Surgery, Guanghua School of Stomatology, Guangdong Provincial Key Laboratory of Stomatology, Sun Yat-sen University, Guangzhou 510055, China; lizhsh26@mail2.sysu.edu.cn (Z.L.); shishw3@mail2.sysu.edu.cn (S.S.); suk@mail.sysu.edu.cn (K.S.); zhenggs@mail.sysu.edu.cn (G.Z.); gaosy@mail2.sysu.edu.cn (S.G.); 2Guangdong Key Laboratory of New Drug Screening, School of Pharmaceutical Sciences, Southern Medical University, Guangzhou 510515, China; lhmlzy@foxmail.com; 3Institute of Polymer Optoelectronic Materials and Devices, State Key Laboratory of Luminescent Materials and Devices, South China University of Technology, Guangzhou 510640, China; zengxuanbbj@foxmail.com (X.Z.); ninghl@scut.edu.cn (H.N.)

**Keywords:** hydrogel, BMSCs recruitment, sequential drug release, osteogenic differentiation, bone regeneration

## Abstract

Bone healing is a complex process that requires the participation of cells and bioactive factors. Stromal derived factor-1 α (SDF-1α) and magnesium ions (Mg^2+^) both are significant bioactive factors for cell recruitment and osteogenesis during bone regeneration. Thus, a bifunctional hydrogel containing a sequential delivery system is fabricated to improve osteogenesis. During sequential delivery of the hydrogel, SDF-1α is predominantly released at the early stage of bone mesenchymal stem cells (BMSCs) recruitment, while Mg^2+^ are constantly delivered at a later stage to improve osteogenic differentiation of recruited cells. In addition, due to the early release of SDF-1α, the hydrogel showed strong BMSCs recruitment and proliferation activity. Mg^2+^ can not only induce up-regulation of osteogenic gene expression in vitro, but also promote bone tissue and angiogenesis in vivo. Taken together, the injection of xanthan gum-polydopamine crosslinked hydrogel co-loading SDF-1α and Mg^2+^ (XPMS hydrogel) provides a novel strategy to repair bone defects.

## 1. Introduction

Bone defects are usually caused by trauma, infection, and tumor resection. Currently, the gold-standard in clinical treatment of bone defects is autologous bone grafting. Although autologous bone grafting can be used to repair a wide range of bone defects, it has the disadvantages of limited bone tissue availability and donor site complications. Nano-hydroxyapatite (nHA) is another bone graft substitute which is usually used to repair alveolar bone defects. However, poor angiogenic efficiency and inadequate osteogenesis are hindering the growth of new bone [1]. With a deeper understanding of bone healing, the number of studies recognizing the importance of cell recruitment in regeneration has increased [2,3,4]. Insufficient responding osteoblast occurs when bone healing process is interrupted, causing obstructed bone formation with competitive growth of fibrous tissues. Therefore, two strategies for establishing a new multi-functional bone tissue substitute are: (1) adapting the shape of the cavity of bone defects to prevent the poor bone healing caused by the fibrous tissues; and (2) rapidly recruiting sufficient quantity of progenitor cells to bone defects, and ordering enhances of cells osteogenic differentiation under differentiation-inducing signals, with the goal of supporting the entire bone regeneration cycle.

Xanthan gum (XG), is a kind of branched polymer extracellular polysaccharide generated with excellent biodegradability and biocompatibility [5,6]. The cross-linking network of hydrogels composed of XG could package and release bioactive molecules, which have been widely used in biomedical fields such as drug release and tissue regeneration [7,8,9]. Stromal derived factor-1α (SDF-1α), also known CXCL12, is a member of the CXC chemokine family that binds specifically to CXCR4 cell membrane receptors [10,11]. Many studies have shown that SDF-1α/CXCR4 axis plays an important role in promoting the recruitment of BMSCs to bone defects through its chemotactic effect [12,13,14]. As a hydrogel formed by natural polymer, XG hydrogels enable to simulate the characteristics of natural extracellular matrices and release the bioactive factors as scaffolds [15]. However, the applications of XG in bone regeneration are less reported due to its insufficient mechanic strength [16]. In the presence of divalent cations, XG could form a weak gel-like structure through its own double helix conformation [17]. At the same time, functional groups on the side chain of XG molecules can be used for chemical modification [18]. Therefore, the mechanical strength and other properties of XG can be adjusted by adding other biological materials or reinforcement materials.

Excitingly, polydopamine (PDA) not only has good biocompatibility, but in particular, it contains rich catechol groups, which can adhere firmly to almost all material surfaces and increase the hydrophilicity of materials, thus contributing to cell adhesion and proliferation [19,20]. In addition, the phenolic hydroxyl moieties in PDA are able to form coordination bonds with metal ions such as Mg^2+^, conferring hydrogels loading capacity and controlled release of Mg^2+^ [21,22]. Meanwhile, our previous study confirmed that the combination of Mg^2+^ and PDA could improve the osteogenic capacity of biomaterial scaffold significantly [23]. In the present study, aiming to simultaneously improved the mechanical properties of the hydrogel and endowed the hydrogel with injectable characteristics, PDA and Mg^2+^ were added to modify XG. Importantly, the addition of Mg^2+^ and PDA to XG hydrogels resulted in the formation of XG-SDF-1α/PDA-Mg^2+^ double crosslinking structure, which would exhibit distinct release kinetics for SDF-1α and Mg^2+^ and enable for controlled release of both bioactive factors sequentially.

In order to achieve rapid and high-quality repairing of bone defects, we developed an intelligent sequential release injectable hydrogel scaffold based on the XG-SDF-1α/PDA-Mg^2+^ double crosslinking structure. SDF-1α would be released at an early stage and recruit BMSCs to defect area, effectively preventing the competitive growth of fibrous tissues. Moreover, long-term angiogenesis and improved osteogenesis would be guaranteed by a sustained release of Mg^2+^ mediated by the chemical interaction with PDA. Finally, the osteogenic functions of hydrogels-HA-Particle scaffolds were successfully performed after being implanted in rat femoral bone defects.

## 2. Materials and Methods

### 2.1. Preparation and Characterization of the SDF-1α/Mg-NPs/Gel (XPMS Hydrogel)

#### 2.1.1. Synthesis of Mg-NPs

A total of 45 mg of dopamine hydrochloride (Sigma-Aldrich, St Louis, MO, USA) and 142.5 mg magnesium chloride (Macklin Biochemical Co., Ltd., Shanghai, China) were completely dissolved in 130 mL of deionized water (DI) under magnetic stirring at room temperature. 20 mL of Tris (2-amino-2-hydroxymethylpropane-1,3-diol, Macklin Biochemical Co., Ltd., Shanghai, China) aqueous solution (22.5 mg/mL) was then quickly added into the above solution. The reaction was allowed to proceed for 8 h. Finally, the product was separated via centrifugation and washed three times with DI water to achieve Mg-NPs.

#### 2.1.2. Synthesis of Xan-CHO

1 g XG (Aladdin, Los Angeles, CA, USA) dissolved in 500 mL of DI water under magnetic stirring at room temperature overnight. Then 634 μL of 107 mg/mL Sodium periodate (Macklin Biochemical Co., Ltd., Shanghai, China) solution was added to reaction for 4 h. Then 24 μL of 1% ethylene glycol was added for 1 h to stop the reaction. Then the product was purified for 24 h with a dialysis bag for molecular weight 3500 D. Finally, the Xan-CHO product was freeze-dried in a freeze dryer (Alpha 1-4 LD plus, Christ, Osterode am Harz, Germany) and stored in a sealed plastic bag at 4 °C for further research.

#### 2.1.3. Synthesis of SDF-1α/Mg-NPs/Gel (XPMS Hydrogel)

Briefly, 100 μg/mL of SDF-1α solution was prepared by completely dissolving 10 μg SDF-1α powder (Peprotech, Rocky Hill, NJ, USA) into 100 μL DI water. In order to obtain SDF-1α/Mg-NPs/gel (XPMS hydrogel), 30 mg of Xan-CHO polymer was mixed with 1 mL of Mg-NPs solution, and 1 μL of SDF-1α solution (i.e., 100 ng SDF-1α) was added. This procedure was conducted under magnetic stirring at room temperature overnight. For the control groups, hydrogels without SDF-1α and Mg-NPs (XA hydrogel), without Mg-NPs (XS hydrogel) and without SDF-1α (XPM hydrogel) were prepared respectively as well.

#### 2.1.4. Characterization

The morphology of Mg-NPs was observed by scanning electron microscope (SEM, Zeiss Merlin Compact, Oberkochen, Germany) and transmission electron microscopy (TEM, FEI Talos F200s, Hillsboro, OR, USA). Hydrodynamic size distribution was evaluated using a Zetasizer Nano ZS90 (Malvern Instruments Ltd., Malvern, Worcestershire, UK) at 25 °C. The surface morphology of XPMS hydrogel was characterized by SEM with energy dispersive spectrometer (SEM-EDS, X-MaxN, Oxford, UK). The mechanical properties of the hydrogels were measured on a rheometer (Physica MCR 301, Anton Poar, Graz, Austria) using a parallel plate with a diameter of 25 mm and a gap of 0.3 mm in an oscillatory mode at 37 °C. Shear stress sweep test of the XPMS hydrogel was conducted with a range of stress from 0 to 100 Pa at a fixed strain of 1%. From the stress sweep, the yield stress of the hydrogel can be obtained. Subsequently, the self-healing property of the XPMS hydrogel was confirmed by the continuous step-torque measurements. Briefly, the XPMS hydrogels were alternately subjected to a torque of 250–370 µN·m to break the hydrogel network and a torque of 4.5–4.6 µN·m to restore the hydrogel network. This process was repeated for two times.

#### 2.1.5. The SDF-1α Releasing from XPMS Hydrogel

The release of SDF-1α from the XPMS hydrogel in vitro was quantified via ELISA. First, the gels were incubated in 15 mL PBS at 37 °C under different conditions. At each time point (1 d, 2 d, 3 d, 4 d, 5 d, 6 d, 7 d), 1 mL PBS was collected and 1 mL fresh PBS was replaced. Finally, the SDF-1α concentrations in the collected PBS were quantified with SDF-1α ELISA kit (Xitang Biotech Company, Shanghai, China) according to the manufacturer’s instructions.

### 2.2. BMSCs Isolation and Culture

BMSCs were isolated and cultured from the femurs of 4 weeks old Sprague-Dawley rats (Animal Research Center, Sun Yat-sen University, Guangzhou, China). Then the isolated cells were then cultured in F12/DMEM (Gibco, CA, USA) supplemented with 10% FBS (Gibco, CA, USA), 100 units/mL streptomycins and penicillin (P/S, Gibco, CA, USA) at 37 °C in an atmosphere of 5% CO_2_. The culture medium was replaced every 3 days. In the following experiments, cells from passages 3 were employed. The BMSCs have been identified by the flow cytometry assay (Appendix A), and the BMSCs are positive for CD90 (BD Biosciences, NY, USA) and CD44 (eBioscience, San Diego, CA, USA), but negative for CD34 (BD Biosciences, NY, USA) and CD45 (eBioscience, San Diego, CA, USA).

### 2.3. The Preparation of Different Hydrogel Extraction

The indirect extraction method was used to evaluate samples’ biocompatibility according to the ISO 10993-12 standard. Briefly, incubate the hydrogels for at least 48 h at 37 °C in F12/DMEM culture medium (containing 10%FBS, 1%P/S). After incubation, each extract solution of blank or drugs-loaded hydrogels samples was prepared by filtering through a 0.22 μm syringe filter (Merck, Darmstadt, Germany) before use. Experimental hydrogel extractions were grouped according to their physical and chemical modification, which divided into five groups: (1) XA group: extraction from pure XG; (2) XP group: extraction from PDA modified XG; (3) XPM group: extraction from Mg^2+^ included PDA modified XG; (4) XS group: extraction from SDF-1α included pure XG; (5) XPMS group: extraction from SDF-1α and Mg^2+^ included PDA modified XG.

### 2.4. The Effect of SDF-1α Included Hydrogels on Recruitment of BMSCs In Vitro

#### 2.4.1. Effect of SDF-1α on Chemotaxis of BMSCs

The recruitment of BMSCs in response to SDF-1α stimulation was investigated using a transwell migration model experiment. Briefly, cells were trypsinized and 200 μL cells suspension was seeded at a cell density of 1.5 × 10^4^ cells per insert into the 8 μm pore transwell chamber (Corning, NY, USA) in a 24-well plate. Then, the chamber was inserted into each well of culture medium containing different concentrations of SDF-1α (0, 50, 100, 200 nM). After 24 h of incubation, the non-migration cells were completely wiped off from the top surface of the membrane using a cotton swab. After being fixed by 4% paraformaldehyde, the cells migrated to the undersurface of the membrane were stained by 0.1% crystal violet. Under the inverted microscope (Zeiss Axio Observer, Oberkochen, Germany), the average numbers of migrated cells were counted in at least five field from each well.

#### 2.4.2. Functional Evaluation of Mg^2+^ and SDF-1α on Chemotaxis of BMSCs

To validate whether SDF-1α and Mg^2+^ have a synergistic effect on the recruitment of BMSCs, the chemotactic responses of BMSCs to SDF-1α (100 ng/mL) in the presence or absence of Mg^2+^ (5 mM) were evaluated using a transwell assay. In a 24-well plate, cells were seeded at a density of 1.5 × 10^4^ cells in the upper chamber. The chamber was then put into each well of culture medium containing SDF-1α and Mg^2+^ alone or both. After 24 h of incubation, the non-migration cells were totally cleaned off the top surface of the membrane using a cotton swab. After being fixed by 4% paraformaldehyde, the cells migrated to the undersurface of the membrane were stained by 0.1% crystal violet. Under the inverted microscope (Zeiss Axio Observer, Oberkochen, Germany), the average numbers of migrated cells were counted in at least five fields from each well.

#### 2.4.3. Cell Recruitment Activity of the XPMS Hydrogels In Vitro

The transwell migration assay was performed to evaluate the chemotactic responses of the BMSCs to the extraction of hydrogels. The upper chamber (pore size of 8 μm, Corning, NY, USA) was seeded with 1.5 × 10^4^ cells. Lower chambers were supplemented with culture medium (10%FBS, 1%P/S) with different hydrogel extractions. The upper chamber was removed after 24 h of incubation, and the cells on the undersurface were stained with 0.1% crystal violet and counted. The average number of migrated cells in each group was calculated using five fields pictured by the inverted microscope (Zeiss Axio Observer, Oberkochen, Germany).

### 2.5. The Effect of Mg^2+^ and Hydrogels on Biocompatibility and Osteoinductivity of BMSCs In Vitro

#### 2.5.1. Proliferation of BMSCs Stimulated with Mg^2+^


The cell counting kit-8 assay (CCK-8, Dojindo, Kumamoto, Japan) was used to investigate the effect of Mg^2+^ on BMSCs. In details, BMSCs were seeded at a density of 1 × 10^3^ per well in 96-well plates and allowed to adhere and spread for 12 h. The BMSCs were then treated for 1, 3, and 5 days with cell culture medium containing F12/DMEM, 10% FBS, 1% (*v*/*v*) penicillin/streptomycin, and various amounts of added Mg^2+^ (0, 2.5, 5, 10, 15, 20 mM). Thereafter, the cells were incubated with the CCK-8 solution for 2 h. A UV-spectrophotometer was used to measure the absorbance at 450 nm. 

#### 2.5.2. Cell Viability of the Hydrogels In Vitro

The biocompatibility of the hydrogels in vitro was investigated using the CCK-8 assay. BMSCs were seeded at a density of 1 × 10^3^ cells per well in 96-well plates and allowed to adhere and spread for 12 h. BMSCs were then treated with different extractions of hydrogels for 1, 3, and 5 days. After that, the cells were incubated with the CCK-8 solution for 2 h. The absorbance was measured at 450 nm using a UV-spectrophotometer. Live/dead assay was used to visualize the morphology and cell spreading of BMSCs under the extraction of hydrogels culturing for 1, 3, 5 days. Briefly, BMSCs were stained by calcein-AM and then pictured by microscope (Zeiss Axio Observer, Oberkochen, Germany) on days 1, 3, 5 after seeding.

#### 2.5.3. The Effect of Mg^2+^ on Osteoinductivity of BMSCs In Vitro

##### ALP Staining and Quantification in BMSCs Treated with Mg^2+^


The alkaline phosphatase (ALP) activity of BMSCs treated with varied Mg^2+^ concentrations was investigated. The BMSCs were cultured in 6-well plates with a growth medium (F12/DMEM, 10%FBS, 1%P/S). After the cells had reached 80% confluence, the medium was changed with osteogenic medium (OM) containing 10% FBS, 1% P/S, 10 × 10^−3^ M β-glycerophosphate (Sigma-Aldrich, St Louis, MO, USA), 50 μg/mL L-ascorbic acid (Sigma-Aldrich, St. Louis, MO, USA), 10 × 10^−9^ M dexamethasone (Sigma-Aldrich, St Louis, MO, USA) and various amounts of added Mg^2+^ (0, 2.5, 5, 10, 15, 20 mM). The cells were rinsed with PBS (PH 7.4, Gibco, CA, USA), fixed in 4% paraformaldehyde, and incubated in p-nitrophenol buffer (Yeason, Shanghai, China) for 30 min according to the manufacturers’ instructions after 7 days of culture. The stained samples were examined with an inverted microscope (Zeiss Axio Observer, Oberkochen, Germany). For the colorimetric measurement of ALP activity, cells were lysed in a lysis buffer and the absorbance at 405 nm was measured by using p-nitrophenol phosphate substrate (Beyotime, Shanghai, China) according to the manufacturer’s protocol.

##### Alizarin Red S Staining of Mineralization in Mg^2+^ Treated BMSCs

Calcium mineralization on the samples was evaluated with the use of Alizarin red S staining (ARS) at day 21. Briefly, cells were seeded in 6-well plates treated with OM (containing F12/DMEM, 10%FBS, 1% P/S, 10 × 10^−3^ M β-glycerophosphate, 50 μg/mL L-ascorbic acid, and 10 × 10^−9^ M dexamethasone) with the following various amounts of added Mg^2+^ (0, 2.5, 5, 10, 15, 20 mM). After culturing for 21 days, cells were immersed in 4% paraformaldehyde for 15 min before being rinsed with PBS. The fixed cells were incubated in 2% ARS solution (Cyagen Bioscience Inc, Guangzhou, China) for 20 min. Finally, after thoroughly rinsing with deionized water, the stained cells were observed using an inverted microscope (Zeiss Axio Observer, Oberkochen, Germany).

#### 2.5.4. The Effect of Hydrogel Extractions on Osteoinductivity of BMSCs In Vitro

##### ALP Staining and Quantification in BMSCs Treated with Hydrogel Extraction

The activity of ALP in BMSCs treated with different hydrogel extractions was investigated. BMSCs were cultured in 6-well plates with a growth medium (F12/DMEM, 10%FBS, 1%P/S). The medium was replaced after 80% confluence with an osteogenic medium extracted from different hydrogels. After 7 days of culture, cells were rinsed with PBS (PH 7.4, Gibco, CA, USA), fixed in 4% paraformaldehyde, and incubated in p-nitrophenol buffer for 30 min according to the manufacturers’ instructions (Yeason, Shanghai, China). The stained samples were examined with the inverted microscope (Zeiss Axio Observer, Oberkochen, Germany). For the colorimetric measurement of ALP activity, cells were lysed in a lysis buffer and the absorbance at 405 nm was determined by using p-nitrophenol phosphate assays (Beyotime, Shanghai, China) according to the protocol of manufacturers’ procedure.

##### Alizarin Red S Staining of Mineralization in Hydrogel Extraction Treated BMSCs

Calcium mineralization on the samples was evaluated with the use of ARS at day 21. Briefly, cells were seeded in 6-well plates treated with OM (containing F12/DMEM, 10%FBS, 1%P/S, 10 × 10^−3^ M β-glycerophosphate, 50 μg/mL L-ascorbic acid, 10 × 10^−9^ M dexamethasone) extracted from different hydrogels. After culturing for 21 days, cells were immersed in 4% paraformaldehyde for 15 min and rinsed with PBS. The fixed cells were incubated in 2% ARS solution (Cyagen Bioscience Inc., Guangzhou, China) for 20 min. Finally, the stained cells were observed using an inverted microscope (Zeiss Axio Observer, Oberkochen, Germany) after thoroughly rinsed with deionized water.

##### Quantitative Real-Time Polymerase Chain Reaction (qRT-PCR)

qRT-PCR analysis was used to evaluate the smaples’ osteogenic capability. Briefly, cells treated with OM (containing F12/DMEM, 10%FBS, 1%P/S, 10 × 10^−3^ M β-glycerophosphate, 50 μg/mL L-ascorbic acid, 10 × 10^−9^ M dexamethasone) extracted from different hydrogels were lysed using the TRIzol reagent (Invitrogen, Carlsbad, CA, USA) after 7 or 14 days of culture. The total RNA concentrations were quantified with a Nanodrop spectrophotometer (Thermo-Fisher, Waltham, MA, USA). The first-strand cDNA was synthesized using RNA reverse transcription kit (Vazyme BioTech, Nanjing, China), and qRT-PCR was performed using a SYBR Green kit (Vazyme BioTech, Nanjing, China) according to the Q5 Quantstudio system protocol. Table 1 shows the primers for the targeted genes.

### 2.6. Osteoinductivity of Different Hydrogels In Vivo

#### 2.6.1. Femur Defect Model of SD Rats

Female Sprague-Dawley rats (SD rats) weighing 210–230 g were randomly assigned to one of five groups: (1) non-treated group; (2) Bio-oss group (Bio-oss^®^ hydroxyapatite granules, HA); (3) XPS group (HA + XPS); (4) XPM group (HA + XPM); (5) XPMS group (HA + XPMS). SD rats were provided by Guangdong Medical Laboratory Animal Center (Sun Yat-sen University, China) and maintained in the center. Animal were fed and handled in accordance with international guidelines on animal welfare and the standards of the Animal Ethical and Welfare Committee of Sun Yat-sen University. The animal experiments were approved by the Animal Ethical and Welfare Committee of Sun Yat-sen University. Briefly, SD rats were anesthetized by 1% sodium pentobarbital and 10% chloral hydrate. The rats’ skin was incised bilaterally in the distal femoral epiphysis, and muscles were dissected bluntly to expose the femoral condyle. Then, using a trephine bur perpendicular to the femur’s shaft axis at a slow speed and irrigated with saline solution to avoid heat necrosis, a 4 mm diameter bone defect was produced. The hydrogels were gently injected to fill the drilled defect with different components. Subsequently, the incision of the soft tissues was then closed, and the rats were given an intramuscular antibiotic injection 3 days after surgery. The rats in this study were sacrificed at 6 and 12 weeks respectively after implantation.

#### 2.6.2. Micro-CT Analysis

The rats were sacrificed at 6 and 12 weeks by administering sodium pentobarbital (90–120 mg/kg), and the bilateral femoral condyles were fixed with formalin (10%) for 7 days. The distal femurs were then scanned using microcomputed tomography (micro-CT, Scanco Medical, Bassersdorf, Switzerland) at 70 kV and 200 μA. Once a region of interest was selected, and the center of the bone defect was reconstructed. Images following 3D construction were used to calculate the bone volume fraction (BV/TV), trabecular number (Tb.N), and trabecular separation (Tb.Sp) in the bone defect for analysis of the bone regeneration process within the defect.

#### 2.6.3. Histological Evaluation

The samples were fixed in formalin and subsequently decalcified with ethylenediaminetetraacetic acid (EDTA, Servicebio, Wuhan, China) for 28 days with gentle shaking, and EDTA changes every 3 days. After decalcification, samples were embedded into the paraffin and serial sections (4 μm thick) were cut from each sample for hematoxylin/eosin (H&E) and Masson’s trichrome staining.

### 2.7. Statistical Analysis

Statistical analyses results were presented as mean ± standard deviation and performed with IBM SPSS V. 25.0 (IBM SPSS). All of the experiments were performed in triplicate. One-way analysis of variance and Tukey’s post hoc test were used to determine the significant differences among the groups. All statistical assessments were performed at a level of 0.05.

## 3. Results

### 3.1. Preparation and Characterization of the SDF-1α/Mg-NPs/Gel (XPMS Hydrogel)

Mg-NPs were first synthesized by the complexation of dopamine and Mg^2+^ under alkaline conditions. The morphology of Mg-NPs was revealed by scanning electron microscope (SEM) and transmission electron microscopy (TEM) observation (Figure 1a,b). Mg-NPs showed a spherical morphology and had an averaged diameter of 70 nm (Figure 1c), and the particle size in the dispersion was found to be stable in PBS conditions after 7 days at 37 °C (Appendix A). The synthesis schematic showed that XPMS hydrogel was synthesized by the crosslinking between the aldehyde groups of modified XG and the amino groups of Mg-NPs through a Schiff base reaction, thus SDF-1α was encapsulated and spread in the gelatinous matrix during the cross-linking process (Figure 1d). The resultant XPMS hydrogel showed black and non-flowing appearance (Figure 1e). As shown in Figure 1f, the SEM image illustrated the irregular and coarse surface structure of the freeze-dried XPMS hydrogel, which favors water ingress and the free drug can diffuse freely from the hydrogel. The overlapped and single element distributions in XPMS hydrogel were shown in Figure 1g. The homogeneous distributions of C, O, N, and Mg indicated that Mg-NPs were uniformly dispersed in the hydrogels. The infrared spectra of the hydrogels showed that the peak of the C=O group at 1732 cm^−1^ became smaller after the introduction of Mg-NPs, indicating the crosslinking between aldehyde groups with the amino groups under the Schiff base reaction (Appendix A). Rheological measurements in the amplitude sweep mode revealed that both the storage modulus G′ and loss modulus G″ showed a plateau at a low shear stress level. The substantially higher of the storage modulus G′ than the loss modulus G″ evidenced the gel state of XPMS hydrogel. As the shear stress increased, the loss modulus G″ decreased while the storage modulus G′ dropped more sharply, and then an intersection of G′ and G″ appeared. This trend indicated the XPMS hydrogel changes from gel to sol state, evidencing its shear thinning property (Figure 1h). As showed in Figure 1I, the hydrogel networks were ruptured by applying a large torque (250–370 µN·m) for 50 s, which led to a liquid-like status (G″ > G′). On the contrary, the hydrogel restored the solid-like mechanics (G′ > G″) immediately once switching the torque to a smaller one (4.5–4.6 µN·m). The G′s could be restored within seconds even after two cycles of network rupture, indicating the quick recovery of the XPMS hydrogel after network failure. Furthermore, the XPMS hydrogel can be remolded into different shapes after injection (Appendix A), and its injectability may be useful for minimally invasive treatments. Then, the long-lasting release of SDF-1α from XPMS hydrogel was evaluated under different conditions (Figure 1j). Compared with SDF-1α/gel (XS hydrogel), the release rate of SDF-1α was remarkably reduced in the presence of XPMS hydrogel attributed to the dense structure by cross-linking between Mg-NPs and XG. XPMS hydrogel showed a much faster SDF-1α release under acidic condition, due to the degradation of Mg-NPs and consequently disruption of the intact gel structure. SDF-1α and Mg^2+^ have different release kinetics, we expected the release rate of Mg^2+^ from XPMS hydrogels was much slower than that of SDF-1α due to the enhanced stability of chemically bonded. As shown in Appendix A, a significantly slower and sustained release of Mg^2+^ was evaluated under different conditions.

### 3.2. Effect of Hydrogels on Recruitment of BMSCs In Vitro

Studies have shown that the activation of the SDF-1α/CXCR4 axis is conducive to the recruiting of BMSCs to the injured site [24]. One of the chemokines, SDF-1α, also known as CXCL12, is secreted around the tissue defects site due to the early inflammation and plays an important role in the recruitment of BMSCs to the defect site. Moreover, SDF-1α has been proved to enhance the BMSCs migration in a dose-dependent manner [25,26]. Figure 2a,b showed that the chemotactic active factor SDF-1α has a significant dose-dependent recruitment effect on BMSCs. 

Many studies have reported the synergistic effect of SDF-1α with other bioactive factors in bone repair [27,28]. We first focused on examining the SDF-1α and Mg^2+^ regulation on BMSCs recruitment. Zhang et al. found that Mg^2+^ also has the potential effect on SMSCs recruitment [29]. Therefore, we speculated cotreatment would improve BMSCs recruitment more than treatment with SDF-1α and Mg^2+^ alone. Figure 2c,d showed that compared to the blank control group, the Mg^2+^ alone group had a significantly higher number of migrated BMSCs, while SDF-1α alone group was higher than that in the Mg^2+^ alone group. These results indicated that the presence of Mg^2+^ improve BMSCs recruitment to some extent, but the significance of SDF-1α in the collaboration with Mg^2+^ also further suggests that SDF-1α delivery was essential for BMSCs recruitment.

SDF-1α encapsulated in crosslinked network and Mg^2+^ complexed by PDA nanoparticles were released in a time-dependent pattern due to the sequential drug release of hydrogels, which may exert a good cell recruitment effect on BMSCs. In order to investigate the effect of sequential drug release hydrogel on cell recruitment, we used hydrogel extract to simulate the effect of hydrogels in situ releasing SDF-1α into the surrounding bone defect microenvironment (Figure 2e). The number of migrated cells in the XPMS group containing SDF-1α and Mg^2+^ increased significantly, showing that the combination of SDF-1α and Mg^2+^ significantly improved the motility and migration of BMSCs. Collectively, the results demonstrated that SDF-1α and Mg^2+^ delivered locally via XPMS hydrogel would have more beneficial biological effects on BMSCs recruitment.

### 3.3. Effect of Hydrogels on Biocompatibility and Osteoinductivity In Vitro

Cell recruitment is the initial step in acquiring endogenous stem cells at a bone defect site, and maximizes the local regeneration ability. Furthermore, effective stem cell differentiation management is required to guarantee that recruited cells differentiate into the desired cell lineages [30,31]. Notably, Mg^2+^ plays an important role in bone mineralization, promoting cell adhesion, proliferation, and osteogenic differentiation [32,33]. Many studies have reported that Mg^2+^ have distinct respective effects at different concentrations. Lin et al. [34] proposed that Mg^2+^ with a concentration gradient of 2.5 mM to 5 mM were optimal for inducing BMSCs osteogenic differentiation, and Wang et al. [35] also proposed that Mg^2+^ with a concentration gradient of 6 mM and 10 mM were best for promoting cell adhesion, cell activity and osteogenic differentiation. Therefore, the concentration of Mg^2+^ is a key factor in modulating the proliferation and osteogenic differentiation of BMSCs.

Figure 3a showed that BMSCs were treated with various concentrations of Mg^2+^ (0, 2.5, 5, 10, 15 and 20 mM) and cell proliferation of BMSCs was evaluated by CCK-8 assays after one, three, and five days. The results showed that 5 mM and 10 mM concentrations of Mg^2+^ significantly promoted the proliferation of BMSCs on day three and day five, while higher concentrations greater than 10 mM inhibited the proliferation of BMSCs. The results were comparable to those previously reported in the literature, indicating that locally high concentrations of Mg^2+^ would impede the mineralization of the bone matrix and damage the process of bone regeneration process [36,37], which was connected to the inhibition of BMSCs proliferation. Therefore, based on our finding, a concentration gradient of 5–10 mM of Mg^2+^ may be a favorable condition for BMSCs proliferation and the subsequent osteogenic differentiation.

Furthermore, SDF-1α has been shown to contribute to the proliferation and the differentiation of BMSCs into osteoblastic cells mediated by Wnt/β-Catenin pathway [38]. Thus, Mg^2+^ and SDF-1α were co-loaded into the hydrogels to investigate the synergistic effect on BMSCs proliferation. Figure 3b showed that, as compared to the XA group, the higher rate of proliferations of BMSCs were observed in XPM and XPMS group on day three, and there was no significant difference in XPM and XPMS group, supporting the notion that Mg^2+^ delivered from hydrogels shows proliferation enhancement toward BMSCs. Meanwhile, morphological investigation was used to confirm the hydrogel extract’s stimulation of BMSC proliferation (Figure 3c). All of the hydrogels showed significant cell density increases and no differences in cell appearance after incubation, which was consistent with the CCK-8 assays. Moreover, BMSCs in XPMS and XPM groups retained spindle shape and showed good proliferation. Collectively, these results suggest that the XPMS hydrogel is highly biocompatible and capable of preserving the survival and proliferation for recruited BMSCs in situ.

The ability of Mg^2+^ to enhance osteogenic differentiation was further investigated by analyzing the activity of ALP, which is an early marker of osteogenesis. The ALP staining of BMSCs was obviously intensified with the increase of Mg^2+^ concentration, and particularly showed the best performance in 5–10 mM (Figure 4a). We also determined ALP activity to provide for quantitative analysis (Figure 4b). ALP expression was enhanced by up to 1.5-fold with 5 mM Mg^2+^, and show no difference with the higher concentration of Mg^2+^. ARS was used to detect calcium deposition as a late osteogenic marker (Figure 4c). The staining of calcium deposition was intensified as the concentration of Mg^2+^ increased up to 5 mM. In contrast, the staining intensity decreased with further increase in Mg^2+^ concentration, suggesting the inhibition of mineralization process. Over all, the osteogenic capacity of Mg^2+^ upregulated with a concentration of 5 mM and significantly promoted osteogenic differentiation of BMSCs, which provide guidance for loading strategy in hydrogels.

Further, to quantify the oestoinductive ability of XPMS hydrogel, ALP staining was performed (Figure 5a). In comparison to the XA group, the ALP staining was intensified in the XPM group and further amplified in the XPMS group. BMSCs in the XPMS and XPM groups exhibited significantly greater ALP activity. Moreover, ARS results (Figure 5b) also demonstrated that the calcium deposition staining in the XPM group was intensified by incorporating Mg^2+^ and exhibited the best performance in XPMS group, which show a similar tendency with the ALP results. The results indicated that the combination of Mg^2+^ and SDF-1α delivered from XPMS hydrogel induced a significantly greater promotion on osteogenic differentiation of BMSCs. Meanwhile, previous studies have shown that SDF-1α stimulation increases the expression of the type I collagen gene (Col I) [39], indicating that SDF-1α regulates BMSCs osteogenic differentiation. Subsequently, qRT-PCR analysis was also used to evaluate the expression of osteogenic genes including COL-I, Runx2, ALP, and OCN (Figure 5c). After culturing for seven days, the mRNA levels of COL-I were upregulated 2.3-fold in XPM group, with XPMS group showing the highest expression (2.95-fold) as compared to XA group. The levels of Runx2 in XPM and XPMS groups were also increased 1.3-fold and 1.2-fold, respectively. The levels of ALP were upregulated 1.4-fold in XPM and XPMS groups. The expression of OCN was significantly enhanced by XPM and XPMS hydrogels on day 14. These findings are consistent with previous report that SDF-1α can participate in osteogenic differentiation of BMSCs by upregulating the expression of COL-I. In addition, Mg^2+^ delivered from the hydrogels plays a significant role in osteogenic differentiation of BMSCs, which suggest that the XPMS hydrogel could create an osteogenic microenvironment by delivering Mg^2+^ and promoting the osteogenic effect of the recruited BMSCs.

### 3.4. Effect of Hydrogels on Osteoinductivity In Vivo

Previous studies have shown that gelatin hydrogels combined with SDF-1α and bone morphogenetic protein (BMP-2) was more effective to induce bone regeneration, which may be attributed to the synergetic effect on intracellular Smad and Erk activation [26]. Other researchers also found that SDF-1α may enhanced recruitment of endothelial progenitor cells (EPCs) by SDF-1α/CXCR4 signaling, which helps to supply nutrient and support bone repair [40]. SDF-1α is not sufficient for in-situ regeneration of critical bone defects, additional osteogenic bioactive factors may also be required. Meanwhile, biomaterial scaffolds containing Mg^2+^ have been shown in numerous investigations to improve osteogenesis [41,42,43]. Moreover, it is worth noting that Mg^2+^ also have a recruiting effect on BMSCs via activating TRPM7/PI3K signaling pathway in recent literature [44]. The therapeutic effect of the hydrogel system containing two bioactive factors was greater than that of the single component system. In order to mimic the process of natural bone healing, XPMS hydrogel possessing the double crosslinking structure was fabricated to control the release sequence of each factor. The crosslinked structure of xanthan gum can prolong the duration of releasing the encapsulated SDF-1α, while the covalent bonding of PDA and Mg^2+^ creates another stable functional microstructure and release profile. Due to the distinct loading mechanisms of SDF-1α and Mg^2+^, SDF-1α is released early around the bone defect and recruits BMSCs and EPCs through SDF-1α/CXCR4 signaling activation, limiting the growth of fibrous tissues. Later, Mg^2+^ are released in a sustained manner, which enhances the proliferation and osteogenic differentiation of recruited cells and accelerates mineralization of bone matrix throughout the bone healing life cycle.

Therefore, to validate the ability of XPMS hydrogel composite scaffolds to promote bone repair in vivo, composite biomaterials of injectable hydrogel containing different bioactive factors were injected into a rat femur defect, micro-CT and histological analysis of femur specimens were performed on 6 weeks and 12 weeks after in situ implantation. As shown in Figure 6a, the non-treated group’s defects displayed a little new bone tissues at the margin, and we observed new bone formation in the Bio-oss group but substantial loss of bone powder in bone defects. The cause is likely to be blood clots forming that was insufficient to provide proper mechanical support on bone powder in the bone defect. In contrast, hydrogel-composite scaffolds showed significant new bone formation at 6 weeks and week 12. At week 6, the XPS and XPMS groups showed appreciable osteogenesis in the defect’s center, forming a “bony bridge” between the bone powder and integrating into patchy. It was confirmed that SDF-1α greatly promote bone defect repair in the early stage by recruiting bone marrow mesenchymal stem cells. The composite hydrogel scaffolds containing the Mg^2+^ (XPM and XPMS groups) showed apparent new bone formation at week 12, suggesting that Mg^2+^ have a substantial role in promoting osteogenesis. 

Further, BV/TV, Tb.N, and Tb.Sp were quantitatively evaluated by micro-CT images. As shown in Figure 6b, composite hydrogel scaffolds containing SDF-1α and Mg^2+^ groups (XPMS group) displayed a noteworthy enhancement of BV/TV at week 12 compared to the non-treated group and the Bio-oss group. Meanwhile, the XPMS group had slightly greater bone parameters than the XPS or XPM groups, suggesting that SDF-1α may have a synergistic effect with Mg^2+^ during bone healing in vivo, which was consistent with our findings in vitro. In contrast, when the XPMS group were compared to the Bio-oss group at week 6, there was no statistically significant difference in Tb.Sp. The trabecular separation of the composite hydrogel scaffold groups (XPS, XPM, and XPMS groups) were considerably lower than the Bio-oss group at 12 weeks after implantation. In conclusion, the composite hydrogel scaffolds may continuously promote the repair of bone defects and enhance osteogenic quality over time, indicating that the injectable bioactive composite hydrogel scaffolds have excellent osteogenesis performance.

The new formation of bone tissues was also evaluated histologically by hematoxylin and eosin (H&E) and Masson’s trichrome staining as shown in Figure 7, and the results were largely consistent with micro-CT analysis. The defects in the non-treated group did not show obvious healing at six weeks, while the Bio-oss group and the composite hydrogel scaffolds group loaded with bioactive molecules (XPM, XPS, and XPMS groups) had more favorable newly woven bone tissue. Moreover, extensive capillaries were observed around the defects in the composite hydrogel scaffolds group. Angiogenesis and osteogenesis are highly correlated in the process of bone development and bone generation according to previous literature [45]. Our findings are in agreement with reports that SDF-1α or Mg^2+^ encapsulated scaffolds have excellent angiogenesis function for bone reconstruction [46,47,48]. Furthermore, at six weeks postoperatively, a large number of osteoblasts were observed in the central area of the defect in the XPMS group as compared to XPS and XPM group, suggesting that the promotion of SDF-1α and Mg^2+^ on the recruitment and proliferation of BMSCs around the defect site.

The histological results at 12 weeks showed that the defects in the non-treated group were filled with fibrous tissues, indicating that wound healing was delayed and showed poor bone healing. In contrast, more woven bone tissues were observed within the defect in the composite hydrogel scaffolds group, and the fibers in the woven bone tissues were more compact and organized than those in the Bio-oss group. Meanwhile, the transition of woven bone tissues into mineralized bone tissues could be found in Masson’s trichrome staining. The XPMS group showed a higher density of mineralized bone tissues growing connectively than the XPS and XPM groups, indicating that SDF-1α and Mg^2+^ released from XPMS hydrogel had successfully accelerated the bone defect repair. Overall, XPMS composite hydrogel scaffolds not only enhance bone marrow mesenchymal stem cells and endothelial progenitor cells recruitment through SDF-1α chemotaxis in early release, but also cooperate with Mg^2+^ released sequentially to induce osteogenesis and angiogenesis, carrying out the entire life cycle of bone restoration.

## 4. Conclusions

In conclusion, a novel injectable bone healing hydrogel was designed with excellent sequential controlled administration of SDF-1α and Mg^2+^. The XPMS hydrogel exhibited good injectable and mechanical properties after introducing Mg-NPs into XG via Schiff base reaction, thus enabling it appropriate for clinical usage. Furthermore, our results suggested that SDF-1α and Mg^2+^ released in a controlled manner from XPMS hydrogels synergistically stimulate osteogenesis, indicating that the underlying mechanism of cell recruitment enhancement at an early stage is involved. This point of view was further supported by a model of femoral bone defects in rats. In our work, the osteogenic capacity of XPMS hydrogels was investigated, and the role of cell recruitment during bone healing was highlighted. These findings may contribute to a new strategy for bone regeneration.

## Figures and Tables

**Figure 1 polymers-14-02872-f001:**
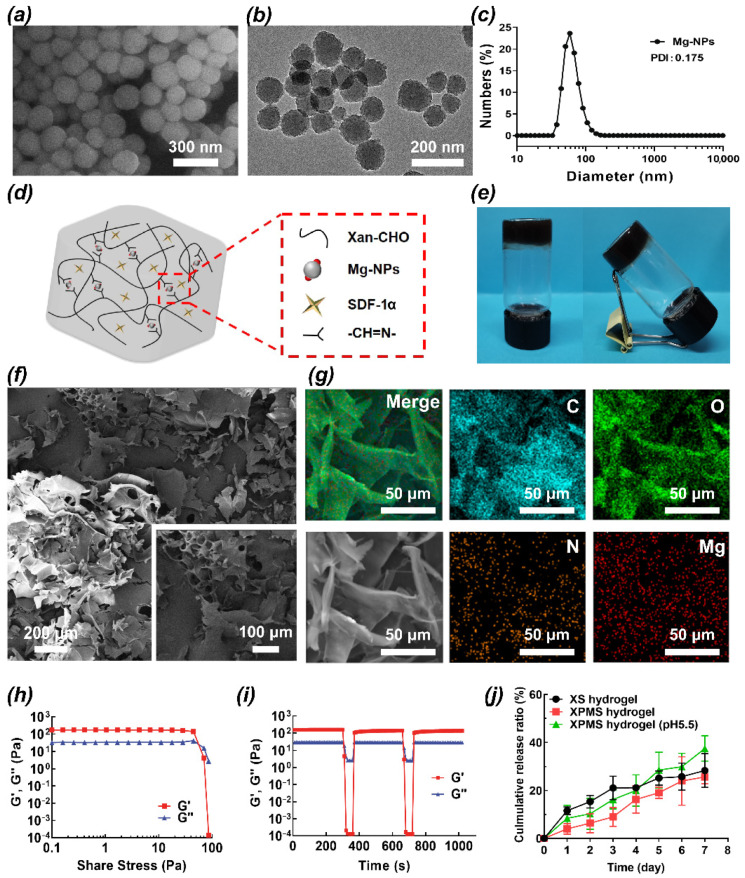
Characterization of XPMS hydrogel. SEM (**a**) and TEM (**b**) images of Mg-NPs. (**c**) Hydrodynamic size distribution profile of Mg-NPs. (**d**) Schematic illustration of the structure of injectable XPMS hydrogel crosslinked by Schiff base bonds. Digital photograph (**e**), SEM images (**f**), and corresponding element mappings (C, O, N, and Mg) (**g**) of XPMS hydrogel. (**h**) Shear stress sweep tests of the XPMS hydrogel at 37 °C. (**i**) Self-healing ability of the XPMS hydrogel evaluated with different torques at room temperature. (**j**) Cumulative release ratio of SDF-1α from the XPMS hydrogel under different conditions in vitro.

**Figure 2 polymers-14-02872-f002:**
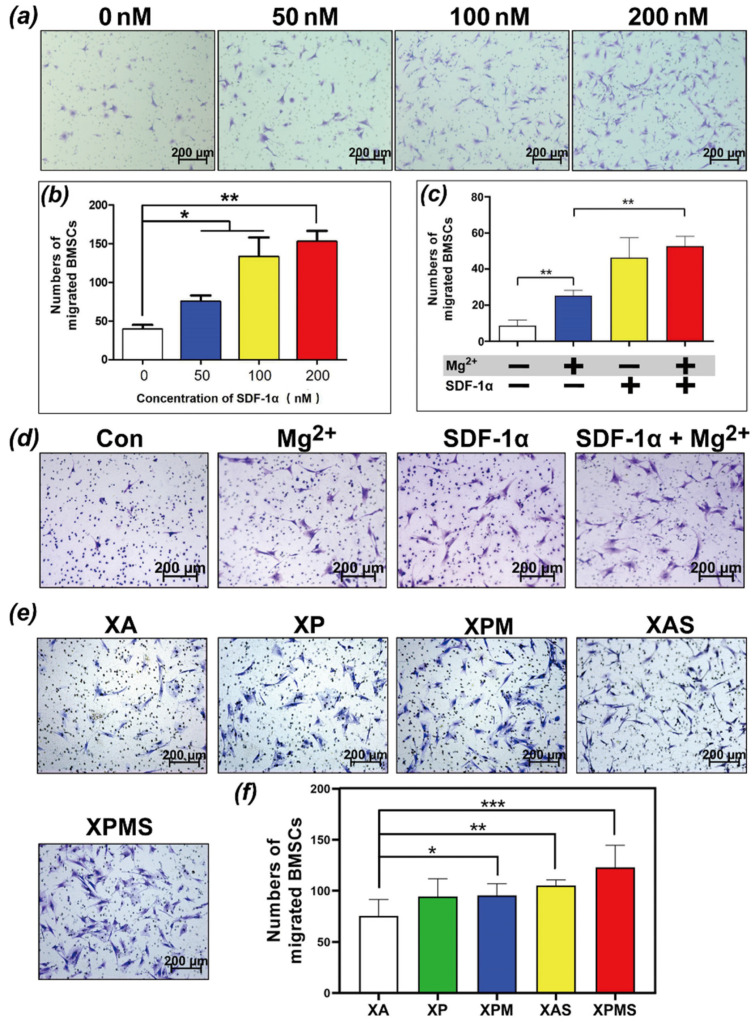
In vitro effect of SDF-1α loaded hydrogels on recruiting BMSCs; (**a**,**b**) the mobility and migration of BMSCs stimulated by different concentrations of SDF-1α were evaluated; (**c**,**d**) the synergistic chemotactic effect of SDF-1α and Mg^2+^ were evaluated; (**e**,**f**) The chemotactic effect of SDF-1α released from hydrogels were evaluated; * *p* < 0.05, ** *p* < 0.01, and *** *p* < 0.001.

**Figure 3 polymers-14-02872-f003:**
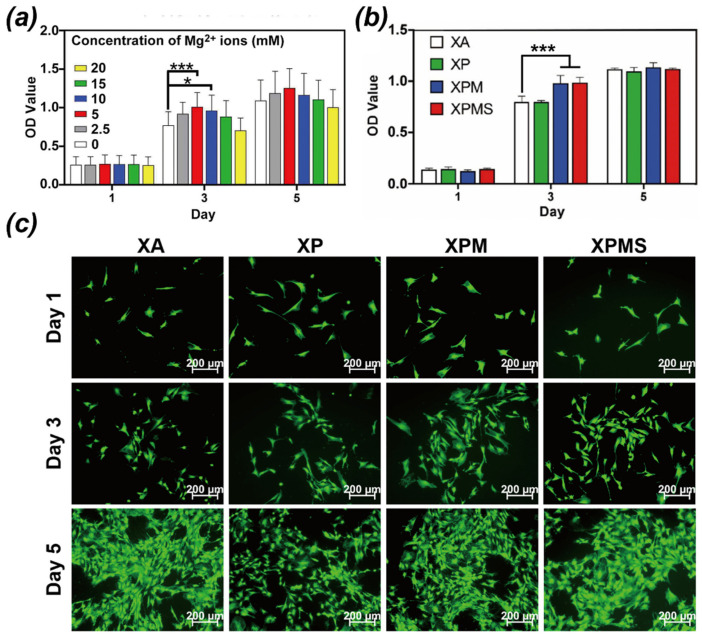
In vitro effect of Mg^2+^ loaded hydrogels on BMSCs proliferation; (**a**) The proliferation of BMSCs treated with different concentrations of Mg^2+^ after one, three, and five days; (**b**) The proliferation of BMSCs in different hydrogel extractions after one, three, and five days; (**c**) Representative fluorescence images of BMSCs in different hydrogels extractions evaluated by Live/Dead staining. * *p* < 0.05, and *** *p* < 0.001.

**Figure 4 polymers-14-02872-f004:**
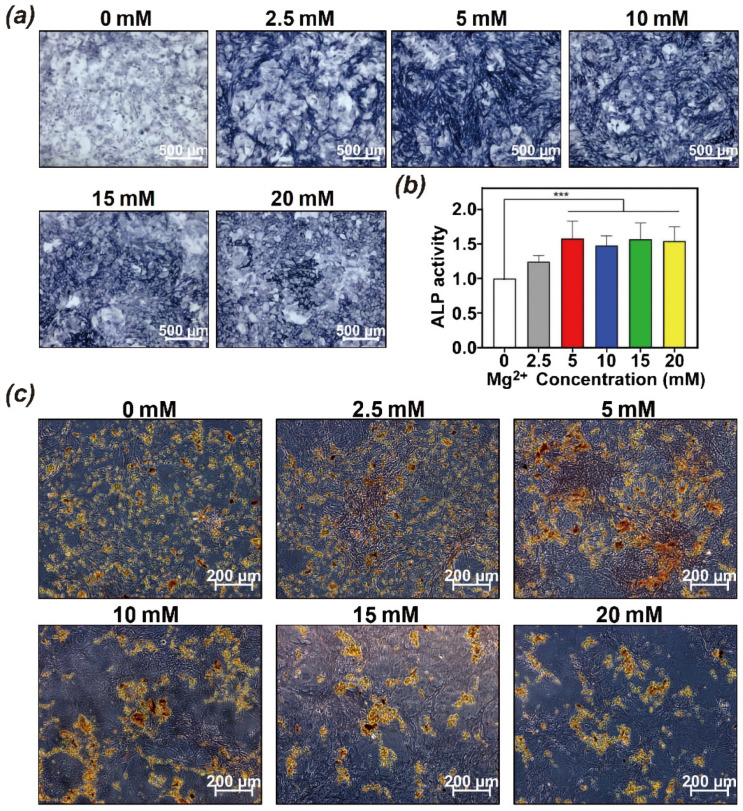
In vitro effect of Mg^2+^ on BMSCs osteogenic differentiation; (**a**) ALP staining and (**b**) quantification of BMSCs exposed to different concentrations of Mg^2+^ for 7d; (**c**) ARS staining of BMSCs exposed to different concentrations of Mg^2+^ for 21d; *** *p* < 0.001.

**Figure 5 polymers-14-02872-f005:**
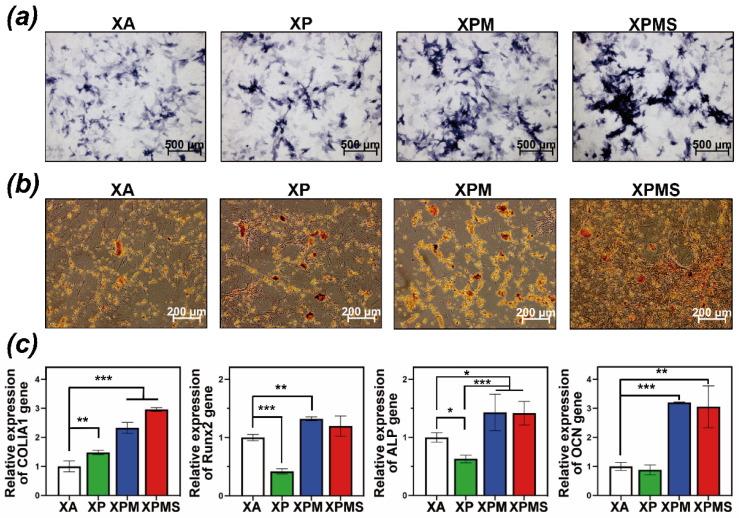
In vitro effect of XPMS hydrogel on BMSCs osteogenic differentiation (**a**) ALP staining of BMSCs cultivated by different hydrogel extractions; (**b**) ARS of BMSCs exposed to different hydrogel extractions; (**c**) relative mRNA expression of osteogenesis related genes (COL-I, Runx2, ALP and OCN) of BMSCs stimulated by different hydrogel extractions. * *p* < 0.05, ** *p* < 0.01, and *** *p* < 0.001.

**Figure 6 polymers-14-02872-f006:**
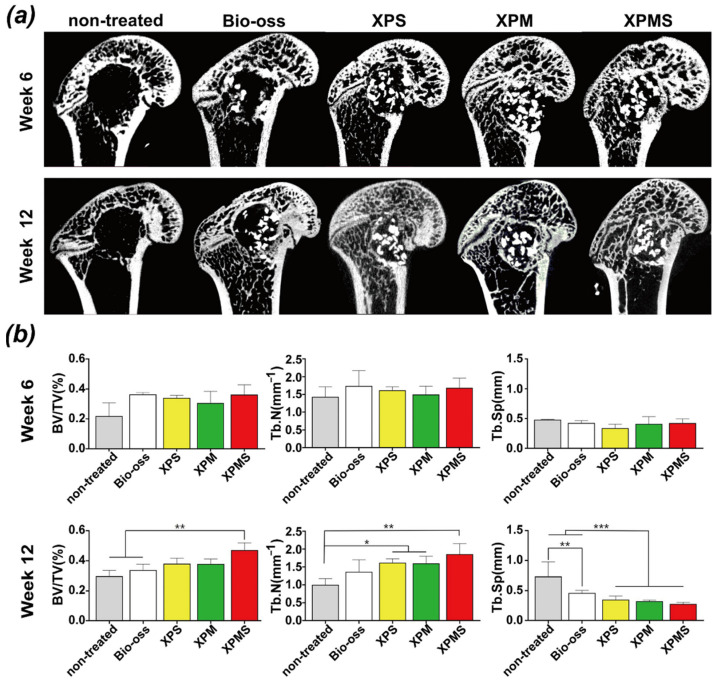
In vivo bone regeneration after 6 and 12 weeks of implantation of XPMS hydrogel on femoral defect. (**a**) Micro-CT images of femoral defects after implantation of different hydrogels scaffold groups for 6 and 12 weeks; (**b**) quantitative evaluations of bone parameters for 6 and 12 weeks, indicating the bone regeneration capability of different hydrogel scaffolds based on histomorphometric micro-CT analysis. * *p* < 0.05, ** *p* < 0.01, and *** *p* < 0.001.

**Figure 7 polymers-14-02872-f007:**
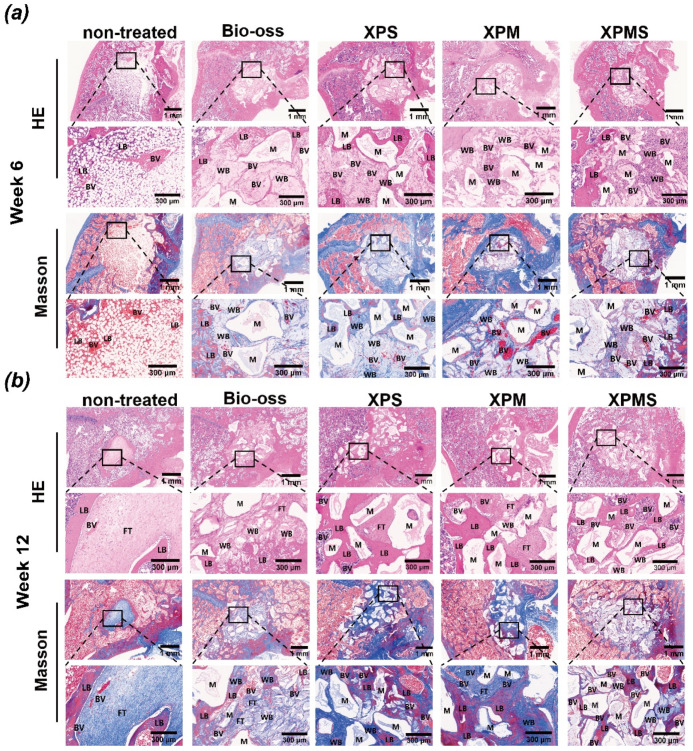
Histological analysis of implantation of XPMS hydrogel on femoral defect after 6 and 12 weeks. (**a**) H&E and Masson’s trichrome-stained histological sections of femoral defects after 6 and (**b**) 12 weeks of implantations; LB, lamellar bone; WB, woven bone; BV, blood vessel; FT, fibrous tissues; M, material.

**Table 1 polymers-14-02872-t001:** Primers Used in qRT-PCR.

Gene	Forward Primer Sequence (5′-3′)	Reverse Primer Sequence (5′-3′)
COL-1	GCTGGCAAGAATGGCGAC	AAGCCACGATGACCCTTTATG
ALP	CAAGGATGCTGGGAAGTCCG	CTCTGGGCGCATCTCATTGT
Runx-2	CAGACCAGCAGCACTCCATA	GCTTCCATCAGCGTCAACAC
OCN	GACAAGTCCCACACAGCAAC	CCGGAGTCTATTCACCACCT
GAPDH	TCTCTGCTCCTCCCTGTTC	ACACCGACCTTCACCATCT

## Data Availability

Not applicable.

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
