# Peer review of "Controlled and Sequential Delivery of Stromal Derived Factor-1 α (SDF-1α) and Magnesium Ions from Bifunctional Hydrogel for Bone Regeneration"

_polymers, 2022, doi:10.3390/polym14142872_

Round 1
Reviewer 1 Report
The analyzed article is important in the field of interest. It brings into discussion the special delivery of SDF-1α and Magnesium ions (Mg2+) as significant bioactive factors for cell recruitment and osteogenesis during bone regeneration.
The manuscript detailed an important experimental plan, results, and discussions as well as the conclusions. May observations and suggestions for the improvement are as follows:
1. Authors are asked to enter an appropriate list of acronyms and their explanations, and these acronyms are to be explained in the text when they are first used. The paper is complex and with a lot of specialized terms.
2. The same observation is true for the abstract section.
3. The title of the article should also include an explanation of SDF-1α used.
4. Scheme one (Scheme 1) could be put into the flow of the manuscript alongside the method of preparation, could also represent a Graphical Abstract, but may not have the position after the Introduction section. There are no references for this Scheme in the text.
5. All the Figures must be with an adequate and uniform resolution, as mentioned in the Guide with Instructions for the authors is mentioned.
5. The Figures put on the non-published material, on the platform are the figures from the manuscript...?
Author Response
Dear Reviewer:
Thank you for expressing interest in our manuscript and we really appreciate the positive remarks concerning our manuscript. We would also like to thank you for their meticulous revision and we are grateful that we are able to largely answer the concerns. The comments are all valuable and very helpful for revising and improving our paper, as well as the important guiding significance to our researches. We have studied comments carefully and have made correction which we hope meet with approval. The responds to the comments are in the attachment, please see the attachment.

Reviewer 2 Report
The manuscript deals with Controlled and sequential delivery of SDF-1α and magnesium 2 ions from bifunctional hydrogel for bone regeneration. This topic is interesting and obtained results are valuable. However, there are some points which decrease the overall quality of the manuscript and which have to be corrected:
1. More references to previous studies should be given in an Introduction part.
2. part 2.1.4 (and some another parts): Several instruments are of unknown source. As to the instructions for authors, manufacturer, type, city and country of origin has to be given for all used instruments.
3. Fig. 1 (h,i,j): Used letters are too small to read them.
4. Fig. 7: Here the letters implemented in images are definitely too small.
All in all, the manuscript is well written and of enough interest but above mentioned corrections have to be done prior its acceptance to the publication.
Author Response
Response to Reviewer 2 Comments
Dear Reviewer:
Thank you for expressing interest in our manuscript (polymers-1782919). We greatly appreciate your positive remarks concerning our manuscript. We also thank you for the constructive and positive comments and suggestions. We have revised the manuscript accordingly. In addition, our point-by-point responses to the comments are listed below this letter.
Point 1: More references to previous studies should be given in an Introduction part.
Response 1: Thanks for your suggestion. We have supplemented more references to the introduction. (See introduction; Lines 41-43; Lines 51- 55; Lines 56-59; Lines 61-64; Lines 70-73)
Point 2: part 2.1.4 (and some another parts): Several instruments are of unknown source. As to the instructions for authors, manufacturer, type, city and country of origin has to be given for all used instruments.
Response 2: We have checked and supplemented the information of instruments we mentioned in the revised manuscript and we have made appropriate corrections. In the revised manuscript, for example, "energy dispersive spectrometer (SEM-EDS, X-MaxN Oxford, UK)" and "the inverted microscope (Zeiss Axio Observer, Germany)" are described.
Point 3: Fig.1 (h,ij):Used letters are too small to read them.
Response 3: According to the suggestion, we have magnified the letters of the Fig.1 (h,i,j).
Point 4: Fig.7:Here the letters implemented in images are definitely too small
Response 4: We sorry for the difficulties in reading the small letters in Fig.7, We have magnified the letters.

Reviewer 3 Report
Overall, the authors goes beyond suggesting a synergism towards the two added compounds (SDF-1alpha and Mg ions) which the results does not provide sufficient statistical data to confirm this. Both compounds are good, but they do not significantly enhance when they are crosslinked together in the hydrogel.
Moreover, the authors also suggest that the compounds are released at different times, i.e., the bioactive factor is firstly released, and then the Mg. However, no results performed by the authors shows this effect. Therefore, this work needs a major revision that address these major points and some others addressed in the comments to the authors section.
Major points
introduction
Lack of references in the introduction, author states a lot of important aspects for many materials without a reference for it.
Mat. and methods
What was the stability of NPs and how the authors avoided agglomeration. The method in question from section 2.1.1 is not convincing that it becomes a stable solution.
Section 2.1.3 does not provide enough information as how the material was formed. For example, it states specific weights but no solvent was used, the only solution provided was with Mg-NPs that was dissolved in a mixture of DI and Tris. How much was produced in order to use in the in vivo.
Isolation of MBSCs were performed, so primary cells were used. Please validate the cell lines. Why the authors used different passages for data analysis. Why not stick with only one passage, i.e., either 3 or 4? This is definetely not standard procedure, please revise this section.
Results
Section 3.1
Images of Mg-NPs does indeed show some agglomeration that may hinder the effect for cyto analysis, and should be mentioned.
Even tough mechanisms for crosslink from xanthan are well described in literature, the authors does not provide data to confirm a double network crosslink as mentioned in text. Only a single diagram is not scientifically accurate and should be better discussed with enough data to provide this argument. Either provide a spectroscopy analysis to fingerprint, that evidences the crosslinked structure or backup with previous works, suggesting that the same reaction occurred in this work.
The authors suggests from Figure 1.F that a porous material is seen, I dont agree with this statement, the cryofracture for SEM analysis usually provide this needle-fracture behaviour and a porosity is not seen in the flat surface of the hydrogel. The internal structure seems to present layers, similar to a chitosan blend structure.
How does the author states that SDF-1alpha was uniform, based on C, O and N from EDS? The polymer may well present these elements and this seems like a far fetch suggestion.
Figure 1.j. I suggest model the release rate, it seems to fit a zero order (XPMS) compared to the XS hydrogel that may fit a Korsmeyer-Peppas equation and should provide enough back up for the explanation in question.
Where is the double release rate of Mg and SDF-1alpha? I only see the bioactive factor. Furthermore, the author states that the factor is first released and then the Mg comes afterwards. Where are the results for this effect?
Section 3.2
As a suggestion for the authors, molecular dynamics and docking should provide enough arguments as well to explain any synergism between Mg and SDF-1alpha. In addition, this synergism would only occur if they were statistically significant different which is not the case. It may have helped but no improvement is seen statistically, please be mindful of the words used in the lines 361-367.
Section 3.3
The authors again states that the Mg significantly increased with SDF-1alpha. It is a true argument for a certain extent and should be explained with caution. in vitro images from Figure 5 clearly presents a visual difference but not statistical data from mRNA expression.
Section 3.5, should be 3.4
Lines 448-453 arguments something not prooved by the authors with no physical evidence. Where is the delayed release of Mg?
Minor comments
abstract
Acronym not explained in abstract - XPMS hydrogel
Introduction
line 46 grammar issue. what does ordering means, and how it enhances differentiation.
Lines 69-71 already previously explained.
mat. and methods.
What does 1d means in line 106.
Add information about where the authors obtained the materials, for example xanthan gum and SDF-1alpha.
Lab analytical balance typically presents an accuracy of 0.001 g, but with an error of ±0.002 g. Please provide the equipment used for such measurement of 100ng.
typo on lines 139, 146, 158, 170, 189, 197, 198, 211, 214 and so on...
What does Zeiss mean on line 165 and 176.
Section 2.6.1.
Injectability properties of the xanthan was not evaluated, how was the material injected.
ISO 10993-5 presents some very distinct differences with the ones proposed by the authors on section 2.3. I suggest the authors specifically state that the procedure was followed by the ISO with some modification, or correct with amendments. For example, the extract is suggested in ISO 10993-12 (using MEM without phenol red and with FBS).
Author Response
Response to Reviewer 3 Comments
Dear Reviewer:
Thank you for your letter and for the reviewer’s comments concerning our manuscript entitled Controlled and sequential delivery of stromal derived factor-1 α (SDF-1α)SDF-1α and magnesium ions from bifunctional hydrogel for bone regeneration. Those comments are valuable and very helpful for revising and improving our paper, as well as the important guiding significance to our researches. According to your nice suggestions, we have made extensive corrections to our previous draft. Please see the attachment.

Round 2
Reviewer 3 Report
The data from supplementary material does indeed answer most of my questions. However, they should be included in the text of the manuscript, supplementary should be completely cited in order for the reader to understand the material and to not be confused. Data from supplementary material should be cited and discussed in the text of the manuscript. They were just included as supplementary and not discussed.
Author Response
Dear Reviewer:
Thank you for your letter and for the reviewer’s comments concerning our manuscript entitled Controlled and sequential delivery of stromal derived factor-1 α (SDF-1α)SDF-1α and magnesium ions from bifunctional hydrogel for bone regeneration. Those comments are valuable and very helpful for revising and improving our paper, as well as the important guiding significance to our researches. According to your nice suggestions, we have made extensive corrections to our previous draft.
Point 1: The data from supplementary material does indeed answer most of my questions. However, they should be included in the text of the manuscript, supplementary should be completely cited in order for the reader to understand the material and to not be confused. Data from supplementary material should be cited and discussed in the text of the manuscript. They were just included as supplementary and not discussed.
Response 1: Thanks for your suggestion. To improve the description of our manuscript, we have rearranged and properly cited the image data from the supplemental material in the revised manuscript s and we have also included a discussion of that as well.
